# Toll-like Receptor 4 Differentially Modulates Cardiac Function in Response to Chronic Exposure to High-Fat Diet and Pressure Overload

**DOI:** 10.3390/nu15245139

**Published:** 2023-12-18

**Authors:** Liping Tian, Mohammad Jarrah, Hussein Herz, Yi Chu, Ying Xu, Yiqun Tang, Jinxiang Yuan, Mohamad Mokadem

**Affiliations:** 1Department of Clinical Pharmacy, School of Basic Medicine and Clinical Pharmacy, China Pharmaceutical University, Nanjing 211198, China; 2Division of Gastroenterology and Hepatology, Department of Internal Medicine, Carver College of Medicine, The University of Iowa, Iowa City, IA 52242, USAyi-chu@uiowa.edu (Y.C.); 3The Collaborative Innovation Center, Jining Medical University, Jining 272067, China; 4Fraternal Order of Eagles Diabetes Research Center, The University of Iowa, Iowa City, IA 52242, USA; 5Obesity Research and Education Initiative, The University of Iowa, Iowa City, IA 52242, USA; 6Iowa City Veterans Affairs Health Care System, Iowa City, IA 52242, USA

**Keywords:** Toll-like receptor, TLR4, cardiac hypertrophy, high-fat diet, pressure overload

## Abstract

Background/Aim: The impact of myocardial stressors such as high-fat diet (HFD) and pressure overload has been extensively studied. Toll-like receptor 4 (TLR4) deficiency has been suggested to have a protective role in response to these stressors, although some conflicting data exist. Furthermore, there is limited information about the role of TLR4 on cardiac remodeling in response to long-term exposure to stressors. This study aims to investigate the effects of TLR4 deficiency on cardiac histology and physiology in response to chronic stressors. Methods: TLR4-deficient (TLR4^−/−^) and wild-type (WT) mice were subjected to either HFD or a normal diet (ND) for 28 weeks. Another group underwent abdominal aortic constriction (AAC) or a sham procedure and was monitored for 12 weeks. Inflammatory markers, histology, and echocardiography were used to assess the effects of these interventions. Results: TLR4^−/−^ mice exhibited reduced cardiac hypertrophy and fibrosis after long-term HFD exposure compared to ND without affecting cardiac function. On the other hand, TLR4 deficiency worsened cardiac function in response to AAC, leading to decreased ejection fraction (EF%) and increased end-systolic volume (ESV). Conclusions: TLR4 deficiency provided protection against HFD-induced myocardial inflammation but impaired hemodynamic cardiac function under pressure overload conditions. These findings highlight the crucial role of TLR4 and its downstream signaling pathway in maintaining cardiac output during physiologic cardiac hypertrophy in response to pressure overload.

## 1. Introduction

Cardiovascular disease has been the leading cause of death in the United States for the past 3 decades [1], and hypertension caused by chronic pressure overload can eventually lead to cardiac remodeling and, ultimately, heart failure [2,3,4]. Obesity is a significant contributing factor to cardiovascular disorders, including heart disease and stroke, which are currently the primary causes of death in both the United States and worldwide [5]. Chronic low-grade inflammation has been well described in the state of obesity at multiple tissue levels but mostly in adipocytes [6,7]. This macrophage-mediated inflammation is orchestrated by the innate immune system predominantly and is key to the development of insulin resistance via the interaction between diet elements and gut microbiome where TLR4 is thought to be a major player [6,7,8]. Toll-like receptors (TLRs) are transmembrane glycoproteins composed of intracellular, extracellular, and transmembrane domains. To date, 10 TLRs have been identified in humans and 12 in mice, with TLR4 being more expressed in the myocardium than other TLRs [9,10,11]. The TLR inflammatory cascade is divided into two main pathways: the MyD88-dependent pathway and the MyD88-independent one. Downstream in the pathways runs interferons (IFNs), regulatory factors (IRFs), and transcription factors such as nuclear factors—κB (NF-κB). Regardless of the pathway, activation of Toll-like receptors induces a pro-inflammatory response via the activation of NF-κB and activator protein-1 (AP-1), which in turn promotes the expression of inflammatory cytokines such as IL-6, IL-1, and TNFα [9].

Pattern recognition receptors (PRRs), including TLRs and others, are important for homeostasis and tissue repair and are part of the innate immune response in the heart following an insult [9,12,13]. PRRs recognize constitutive and conserved pathogen-associated molecular patterns (PAMPs) and endogenous markers called damage-associated molecular patterns (DAMPs) as a mechanism of action [9,14]. Hu et al. showed that high-fat diet (HFD) feeding in mice led to various cardiometabolic derangements starting from overt obesity to insulin resistance, oxidative stress, and glucose intolerance, reaching cardiac contractile dysfunction [15]. Interestingly, TLR4-deficient mice alleviated these effects, exerting a protective effect that could be attributed to autophagy and inflammatory response damping [15]. The TLR4 pathway has also been associated with heart failure and implicated in cardiac remodeling and hypertrophy [16,17,18,19]. Additionally, NF-κB activation seems to be necessary for the development of cardiac hypertrophy as a response to aortic constriction (AC), as inhibiting it abates the hypertrophy [20]. Moreover, mice subjected to pressure overload exhibited elevated mRNA levels of TLR4 in their hearts and contributed to cardiac remodeling, along with high TLR2 expression [16]. Research has demonstrated that HFD and abdominal aortic constriction (AAC) can have various impacts on different organs, including the heart. The development of cardiac hypertrophy as a result of these stresses may be the result of the interaction of multiple pathways, such as TLR4 [15,20,21]. While TLR4 deficiency has been shown to offer cardioprotective benefits when exposed to HFD [15] and AAC [16,17,18,20], other studies showed the exact opposite effect as a response to pressure overload. In such cases, this deficiency led to worsening of the injury [22]. Furthermore, some studies that examined the effects of TLR4 on cardiac hypertrophy did not assess dynamic cardiac function [20,21] or examined it but only after a few days of aortic constriction surgery [17]. Thus, the objective of this study is to investigate the role of TLR4 signaling in modulating cardiac histology and dynamic physiology after chronic exposure to HFD or AAC. Ultimately, the goal is to try to further clarify the previously observed results and potentially help develop future therapeutic interventions for these conditions.

## 2. Material and Methods

### 2.1. Animal Study

TLR4^−/−^ mice (The Jackson Laboratory, JAX 029015) of the C57BL/6J genetic background and wild-type (WT) C57BL/6J (JAX 000664) mice were used. All animal care, experiments, and procedures were approved by the University of Iowa Animal Care and Use Committee (IACUC Protocol # 0092335-005) and under the VA Animal Component of Research Protocol (ACORP, Protocol # 22910010). The study was conducted according to the guidelines of the Declaration of Helsinki and approved by the IACUC of the University of Iowa and the Iowa City VA Health Care System ACORP committee. Six-week-old WT and TLR4^−/−^ male mice were randomly divided into the following groups: WT fed with a normal chow diet (WT ND) (59% carbohydrates, 23% protein, 18% fat, Teklad 7913), WT fed with a high-fat diet (WT HFD) (60% calories from fat, D12492, Research Diets, Inc., New Brunswick, NJ, USA), TLR4^−/−^ fed with a normal diet (TLR4^−/−^ ND), and TLR4^−/−^ fed with a high-fat diet (TLR4^−/−^ HFD). Feeding lasted for 28 weeks. Body weight was measured weekly. Echocardiography was performed by a single experienced echocardiologist who was blinded to the identity of the mice. AAC requires simple surgical techniques, and the results are highly reproducible. All animals were given a weight-based dose of prophylactic antibiotics (enrofloxacin, 5 mg/kg) and analgesics (buprenorphine, 0.1 mg/kg) for pain management before laparotomy. TLR4^−/−^ and WT male mice, 8–10 weeks old and weighing 20–25 g, were used for AAC surgery. After fasting overnight (maximum of 16 h), mice were anesthetized with 1.5% isoflurane and 0.8 L/min O_2_. A 6.0 silk suture was placed under the abdominal aorta, two loose knots were tied, and a 27 G (0.4 mm) blunt needle was placed parallel to the abdominal aorta. The first knot was quickly tied against the needle, followed by the second knot. The needle was promptly removed to produce a constriction of 0.4 mm in diameter. In sham mice, all procedures were identical except for the banding of the aorta step. Body weight was measured weekly. Echocardiography was performed after 60 days of operation to evaluate the heart function.

### 2.2. Echocardiography

Trans-thoracic two-dimension(2D) Doppler-echocardiography was used for cardiac evaluation in mice. Mice were lightly anesthetized with Midazolam (2 ng/mL, SC). Parasternal long- and short-axis views were obtained using echocardiography (Vevo 2100; VisualSonics, Toronto, ON, Canada) to assess left ventricle (LV) mass, volumes, and systolic function [23].

### 2.3. Histology

Hearts were harvested and embedded in an optimal cutting temperature compound (OCT) compound in a cryomold. Sections 10 µm thick were cut with a cryostat. Sections were stained with hematoxylin & eosin (H&E) and Masson’s stains to evaluate inflammation and collagen, respectively. Myocyte cross-sectional area (MCS) quantification and collagen volume fraction (CVF) were quantified using computer-assisted image analysis software FIJI-Image J Plus (ImajeJ2 or version 2).

### 2.4. Quantitative Reverse Transcription-Real-Time Polymerase Chain Reaction (RT-qPCR)

Total RNA was extracted from the heart with TriZol (Thermo Fisher, Waltham, MA, USA) and RNeasy Mini Kit (QIAGEN, Hilden, Germany) according to the manufacturer’s instructions. Total RNA was reverse transcribed to cDNA using cDNA High-Capacity Reverse Transcription Kit (Applied Biosystems, Waltham, MA, USA, Thermo Fisher, Waltham, MA, USA) and TaqMan Fast Universal PCR Master Mix (2×) (Applied Biosystems, Waltham, MA, USA, Thermo Fisher, Waltham, MA, USA) was used for RT-qPCR in a qPCR instrument (QuantStudio 3, Applied Biosystems, Waltham, MA, USA). TaqMan primers/probes were purchased from Integrated DNA Technologies: NFkbib (Mm.PT.58.8646984), p65 (Mm.PT.58.29633634), Tlr4 (Mm.PT.58.41643680), Smad7 (Mm.PT.58.6640883), Acta2 (α-smooth muscle actin, Mm.PT.58.16320644), Tgfb (Mm.PT.58.8169936), p65 (Mm.PT.58.29633634), Ikbb (Mm.PT.58.8646984), caspase-3 (Mm.PT.58.13460531), and RplpO (Mm.PT.58.43894205). Rplp0 as a reference gene (with HEX dye) and gene of interest (FAM dye) were quantified in a single reaction with identical RT input. Relative expression levels were determined using the 2^−∆∆CT^ method, as described previously [24].

### 2.5. Statistical Analysis

All data were shown as mean ± SEM. Student’s *t*-test was used to compare two groups, and one-way analysis of variance (ANOVA) followed by Tukey–Kramer post hoc analysis was used to compare three or more groups. *p* value less than 0.05 was considered statistically significant. Statistical analyses were performed using GraphPad Prism 9.0.

## 3. Results

### 3.1. TLR4 Deficiency Protects against HFD-Induced Weight Gain, Cardiac Hypertrophy, and NF-κB-Mediated Inflammation

As expected, HFD causes weight gain in TLR4^−/−^ as well as wild-type (WT) mice, but the effect was significant in TLR4-deficient mice compared to their WT controls (Figure 1A,B). Additionally, TLR4-deficient mice on HFD exhibited lower heart weight (even after adjusting for total body weight) and lower heart weight/tibia length ratio when compared to WT mice (Figure 1C,D). Interestingly, cardiac TLR4 expression was not altered by chronic HFD exposure in WT controls (Figure 2A). The mRNA levels of SMAD7, α-SMA, in addition to p65 and IκBb subcellular components of NF-κB—a downstream regulator of TLR4 signaling—were all decreased in TLR4-deficient mice exposed to HFD, indicating reduced inflammation (Figure 2B–E). On the other hand, caspase-3 levels were not different in WT and TLR4-deficient mice on either ND or HFD (Figure 2F). These findings suggest that the inflammatory cascade observed in WT mice on HFD is not likely TLR4-mediated.

### 3.2. Despite Protection against HFD-Induced Myocardial Hypertrophy and Fibrosis, TLR4 Deficiency Does Not Alter Hemodynamic Cardiac Function

Using immunohistochemistry to evaluate the effect of high-fat diet on cardiac microanatomy, we found that the myocyte cross-sectional area (MCSA) and collagen volume fraction (CVF) were notably lower in TLR4^−/−^ mice compared to their WT controls on HFD but not on ND (Figure 3A–D). The hemodynamic cardiac function was assessed using 2D echocardiography, which revealed that neither TLR4 deficiency nor HFD interventions had any impact on ejection fraction (EF), left ventricular (LV) thickness, end-systolic volume (ESV), or end-diastolic volume (EDV) (Figure 3E–H). These findings suggest that “TLR4 deficiency” offers protection to cardiac tissue by reducing fibrosis and inflammatory response in mice but does not confer any observable physiological benefits in the absence of additional injuries or stressors.

### 3.3. Effect of Abdominal Aortic Constriction (AAC) on NF-κB-Mediated Inflammation

AAC was found to have different effects on various inflammatory markers in mice. First, we confirmed that AAC caused elevated cardiac TLR4 levels in WT mice after 12 weeks after intervention (Figure 4A). Furthermore, SMAD7, α-SMA, p65, and IκBb levels remained consistent across all interventions (Figure 4B–E). This suggests that chronic pressure overload induced by AAC increases cardiac TLR4 levels but has minimal effects on NF-κB-mediated inflammation. On the other hand, caspase-3 levels were elevated in WT after 12 weeks of AAC, a finding that was absent in TLR4^−/−^ mice (Figure 4F).

### 3.4. TLR4 Deficiency Worsens Hemodynamic Cardiac Function in Response to AAC

Using 2D in vivo echocardiography, we observed that the left ventricle thickness increased significantly in WT mice in response to AAC but only slightly in TLR4^−/−^ AAC mice, which was not statistically significant (Figure 5A,B). The heart weight/tibia length ratio increased significantly in response to constriction when compared to sham intervention (Figure 5C). The EF% decreased in TLR4^−/−^ AAC mice compared to TLR4^−/−^ sham mice and WT AAC mice (Figure 5D). While AAC had no major effects on EDV, ESV, or HR in WT mice, it induced an increase in ESV in TLR4^−/−^ compared to their sham counterparts (Figure 5E–G). Furthermore, we observed an increase in cardiac septum cellularity, thickness, and fibrosis (using H&E and Masson’s staining) in response to AAC across both genotypes (Figure 6) but more in the TLR4^−/−^ group compared to WT controls, which may relate to the observed decline in cardiac function. TLR4 had a protective effect on cardiac function in WT mice, whereas TLR4 deficiency resulted in a mild decline in physiological heart function post-AAC (i.e., under constrictive conditions).

## 4. Discussion

Mechanical, metabolic, or genetic stresses can eventually cause adaptive responses like cardiac hypertrophy to match the increased demands. However, if these stresses persist, these adaptations can become maladaptive and ultimately lead to cardiac dysfunction, potentially resulting in heart failure [25,26,27]. A previous study showed that cardiac hypertrophy induced by HFD alone is not enough to cause heart failure, as demonstrated by dynamic echocardiography [28]. In our study, and to ensure a comprehensive evaluation of cardiac function, we utilized imaging techniques alongside histological analysis that confirmed this finding.

We found that TLR4^−/−^ mice were protected from cardiac hypertrophy when exposed to HFD in comparison to their WT counterparts. This protective effect was evident via histological examination, as the TLR4^−/−^ group exhibited less hypertrophy and fibrosis than the WT group. The TLR4 pathway downstream markers, namely p65 of NF-κB and IκBb, were significantly lower in the TLR4 deficient group than the control group, which could potentially account for the decreased fibrosis and hypertrophy in the heart. In addition, reductions in SMAD markers strongly correlated with cardiac protection and reduced inflammation. Post-AAC assessment showed that in WT, there was significant hypertrophy evident by echocardiography and an increase in LV thickness, which was not observed in TLR4^−/−^ mice. TLR4 deficiency has previously been shown to cause less cardiac hypertrophy after myocardial infarction (MI) or pressure overload induced by transverse aortic constriction, but the studies did not address physiologic cardiac function in vivo. Furthermore, ablation of downstream metabolites in the TLR pathway in mice, such as NF-κB, was also shown to be protective against post-MI ventricular dilation and fibrosis and preserves LV function [16,17,29,30,31].

To evaluate cardiac function in the setting of TLR4 deficiency, we used dynamic 2 D echocardiogram in TLR4^−/−^ mice measuring EF%, ESV, and EDV post-AAC. Surprisingly, EF% was significantly reduced in TLR4^−/−^ AAC compared to TLR4^−/−^ sham and WT AAC, while ESV was significantly increased in TLR4^−/−^ AAC compared to the same groups. While maintaining a normal EF%, the TLR4^−/−^ mice exhibit an impaired cardiac function post-AAC; this could be an initial first hit to the cardiac cells, after which severe injury and dysfunction may develop. This was also evident histologically via the increased Masson’s staining in the hearts of TLR4^−/−^ AAC mice. In relation to the TLR cascade, p65 and IκBb levels were not significantly different between the groups, and the other inflammatory markers could not further explain the observed changes. TLR4 was increased in the WT mice that underwent AAC but did not yield any cardiac abnormalities evident by histology and echocardiography. Therefore, the post-AAC cardiac effect cannot be attributed to the TLR4 pathway, suggesting the involvement of an alternative pathway. Animals lacking TLR4 are relatively protected from HFD induced obesity and diabetes, whereas animals lacking TLR5, on the other hand, have an increased risk of obesity [8,32,33,34,35,36,37]. Additionally, doxorubicin-induced systemic inflammation is thought to result from activation of the TLR4 inflammatory pathway as well as endotoxin leakage, contributing to the toxic side effects of this chemotherapeutic agent [38]. The full roles and response mechanisms of TLRs are still not fully elucidated, requiring further research and cautious implementation of our understandings and observations. TLR2 and TLR4 have been shown to play a role in the pathogenesis of dilated cardiomyopathy. Surprisingly, Ma et al. showed that after blockade of the TLR4 pathway, inflammation, cardiac dysfunction, and fibrosis increased rather than decreased, and only after TLR2 blockade did we observe a decrease in the extent of cardiac dysfunction, fibrosis, and myocardial inflammation [22]. This clearly demonstrates the complexity of the TLR system and the differential response of each component, depending on the context of the study and experiment, thus requiring extensive research to try to explain the outcomes related to the TLR system and/or the other possible pathways.

TLR4 activation plays a crucial role in both negative and positive aspects of cardiac function. On one hand, the activation of the TLR4 signaling cascade can lead to detrimental effects on cardiac function by promoting inflammation and cardiac muscle inflammation. Excessive TLR4 activation has been associated with chronic inflammation and autoimmune diseases [39]. Activation of TLR4 in cardiac cells has been shown to initiate the release of inflammatory cytokines, contributing to cardiac hypertrophy [40]. Chronic infusion of lipopolysaccharide derived from Porphyromonas gingivalis (PG-LPS) has been found to induce cardiac fibrosis via TLR4 activation, causing decreased left ventricular ejection function and increased apoptotic myocytes, fibrosis, and oxidative DNA damage [39]. On the other hand, TLR4 activation also has a positive role in inducing cardiac hypertrophy when needed. In response to stress or injury, TLR4 can initiate an adaptive response, leading to cardiac hypertrophy. This hypertrophy is a compensatory mechanism to increase cardiac output and maintain cardiac function during times of increased demand or injury [40]. The activation of TLR4 signaling in cardiac cells triggers intracellular signaling pathways, immune cell infiltration, and the secretion of inflammatory mediators, which contribute to cardiac hypertrophy [39,40]. Interestingly, we found that caspase-3 was not elevated after chronic exposure to HFD but was significantly increased in cardiomyocytes of WT or TLR4^−/−^ animals after long exposure to AAC. Intriguingly, a previous report has confirmed that activation of caspase-3 was sufficient to cause cardiomyocyte hypertrophy in adult Sprague Dawley rats [41]. This further reinforces our findings where AAC induced LV hypertrophy and elevation in caspase-3 in WT but not TLR4-deficient mice.

Further research is needed to fully understand the balance between the negative effects of TLR4 signaling activation on cardiac function and the positive role of TLR4 activation in inducing cardiac hypertrophy.

Our findings suggest that TLR4 signaling may play a key role in chronic HFD-induced cardiac hypertrophy and inflammation but does not by itself modulate dynamic cardiac function. This protective role of TLR4 deficiency on chronic HFD-induced inflammation seems to be mediated via the NF-ĸB-related pathway. On the other hand, TLR4 deficiency was detrimental to cardiac function after chronic pressure overload secondary to AAC. This effect of TLR4 on cardiac remodeling does not seem to be mediated by an inflammatory cascade, at least not in the case of NF-ĸB or SMAD. TLR4 deficiency could play this protective effect via the AMPK and mTOR pathways, as a result of promoting autophagy [15], in addition to its traditional MyD88 dependent and independent pathways [9]. In addition to the TLR4 pathway, the PI3K/Akt/mTOR signaling pathway has also been shown to play a role in cardiac hypertrophy caused by pressure overload and blocking it almost diminished hypertrophy. When the PI3K/Akt/mTOR pathway was inhibited, a more prominent decrease in cardiac hypertrophy, NF-κB, and IκBb was noted, providing evidence that TLR4 downstream metabolites can be affected by different TLR pathways which could be contributing to cardiac hypertrophy. However, this study was performed on mice with a C.C3H background, not the C57BL/6J model used in most other experiments [21].

The complex involvement of multiple pathways, along with our limited knowledge of them, highlights the need for caution when attempting to translate these findings into therapeutic strategies dealing with cardiac hypertrophy and fibrosis in response to various stresses and stimuli. Moreover, understanding the different models is a crucial challenge in unraveling the adaptive responses of cardiac remodeling and function to pressure overload. We show that TLR4 was able to confer cardiac protectivity against HFD-induced cardiac changes, whereas it was associated with decreased functionality of the heart in response to AAC. These findings suggest a differential role of TLR4 in regulating myocardial cell growth in response to inflammation vs. physiology pressure overload. Therefore, we suggest a careful examination of the etiology of cardiac hypertrophy (or failure) when evaluating TLR4 signaling as a target for possible future cardiac therapies.

## 5. Conclusions

The study explores the role of TLR4 (Toll-like receptor 4) in cardiac responses to various stresses. While TLR4 deficiency protects mice from cardiac hypertrophy induced by a high-fat diet (HFD), it surprisingly leads to impaired cardiac function after chronic pressure overload due to aortic constriction (AAC). The protective effect against HFD-induced inflammation appears to be mediated via the NF-κB-related pathway. In contrast, TLR4 deficiency’s impact on AAC-induced cardiac remodeling suggests involvement in pathways beyond NF-κB or SMAD, potentially implicating AMPK and mTOR pathways. The complexity of TLR4 signaling, along with interactions with other pathways, underscores the challenges in translating these findings into therapeutic strategies for cardiac hypertrophy and fibrosis.

## Figures and Tables

**Figure 1 nutrients-15-05139-f001:**
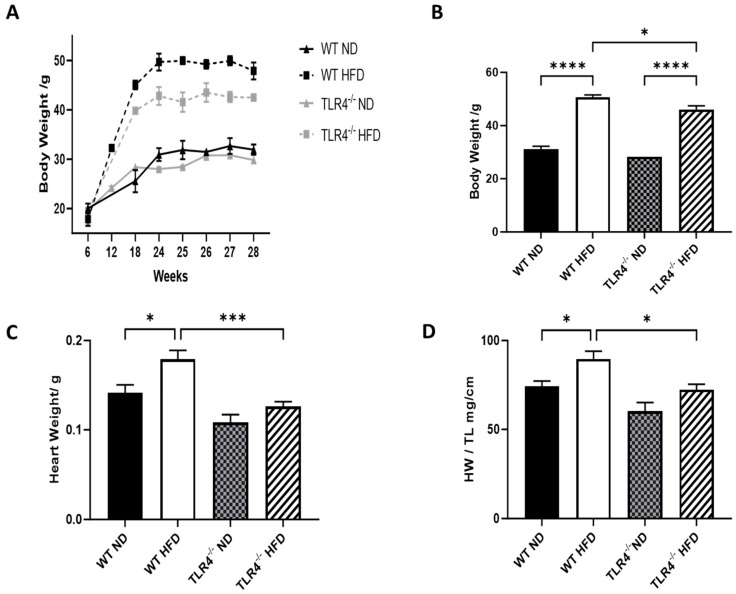
TLR4 deficiency attenuates high-fat diet-induced weight gain and cardiac hypertrophy. (**A**,**B**) Body weights in grams (g), (**C**) heart weights, and (**D**) heart weight/tibia length (HWTL) at the time of tissue harvest in TLR4^−/−^ mice and WT controls I response to ND and HFD. *n* = 6 or 10. Statistical significances are denoted with asterisks as follows: * *p* < 0.05, *** *p* < 0.001, **** *p* < 0.0001.

**Figure 2 nutrients-15-05139-f002:**
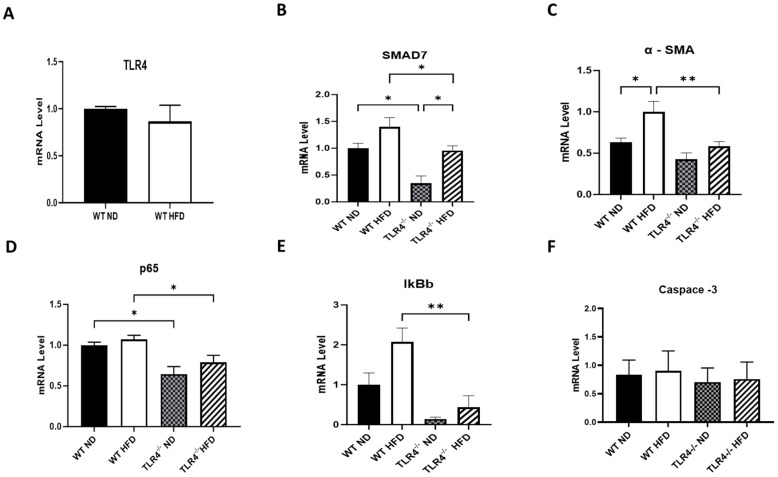
HFD-induced activation of SMADS and NF-κB signaling pathway is attenuated in TLR4 deficient mice. Cardiac mRNA levels of (**A**) TLR4, (**B**) SMAD7, (**C**) α-SMA, (**D**) p65, (**E**) IkBb, and (**F**) caspase-3 in TLR4^−/−^ mice and WT controls after 28 weeks exposure to either ND or HFD. *n* = 4–6. Statistical significances are denoted with asterisks as follows: * *p* < 0.05, ** *p* < 0.01.

**Figure 3 nutrients-15-05139-f003:**
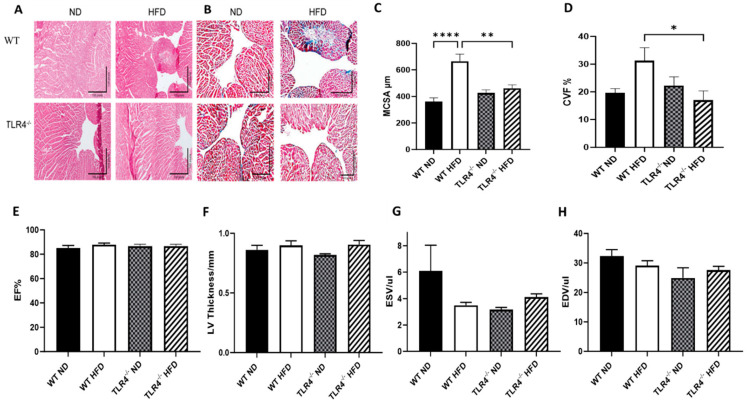
TLR4 deficiency attenuates HFD-induced myocardial inflammation and fibrosis but has no effect on cardiac hemodynamic function. Myocardial (**A**) H&E staining images (10×), (**B**) Masson’s staining images for collagen (5×), (**C**) myocyte cross-sectional area (MCSA) quantification in heart (45 myocytes/mice), (**D**) collagen volume fraction (CVF) in cardiac heart tissue expressed as collagen area/total myocardial area*100% in TLR4^−/−^ (**E**) ejection fraction (EF), (**F**) LV thickness, (**G**) end systolic volume (ESV), (**H**) end diastolic volume (EDV). in TLR4^−/−^ and Wild-type (WT) controls on ND and HFD. *n* = 5–6. Statistical significances are denoted with asterisks as follows: * *p* < 0.05, ** *p* ≤ 0.01, **** *p* < 0.001.

**Figure 4 nutrients-15-05139-f004:**
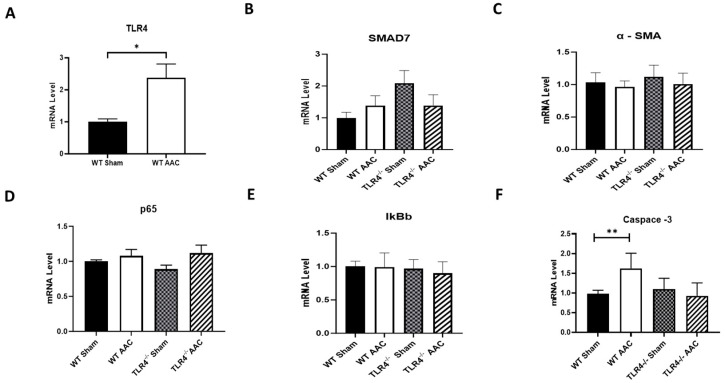
Effect of abdominal aortic constriction (AAC) on TLR4-dependent inflammatory cascade. Cardiac mRNA levels of (**A**) TLR4, (**B**) SMAD7, (**C**) α-SMA, (**D**) p65, (**E**) IkBb, and (**F**) caspase-3 in TLR4^−/−^ mice and WT controls after 12 weeks exposure to either ND or HFD. *n* = 4–6. Statistical significances are denoted with asterisks as follows: * *p* < 0.05, ** *p* < 0.01.

**Figure 5 nutrients-15-05139-f005:**
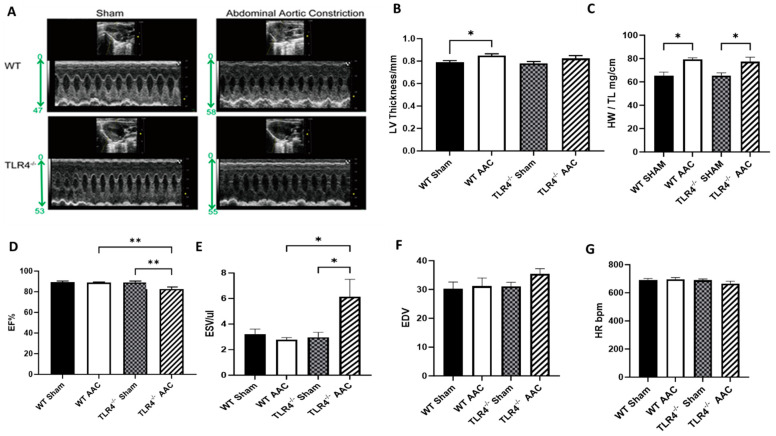
TLR4 deficient mice are protected from cardiac hypertrophy induced by AAC. (**A**) Representative echocardiography images 60 days after operation, (**B**) left ventricle (LV) thickness, (**C**) quantitative analysis of heart weight/tibia length (mg/cm) 84 days post-intervention, (**D**) ejection fraction (EF), (**E**) end-systolic volume (ESV), (**F**) end-diastolic volume (EDV), and (**G**) heart rate (HR) in TLR4^−/−^ mice and WT controls in response to AAC. *n* = 6–10. Statistical significances are denoted with asterisks as follows: * *p* < 0.05, ** *p* ≤ 0.01.

**Figure 6 nutrients-15-05139-f006:**
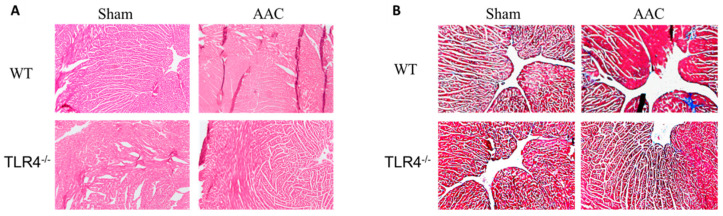
(**A**) H&E staining images (10×) and (**B**) Masson’s staining images (5×) of cardiomyocytes in TLR4^−/−^ and WT controls in response to AAC.

## Data Availability

All data obtained were published and will be made accessible to all viewers and researchers.

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
