# Peer review of "Toll-like Receptor 4 Differentially Modulates Cardiac Function in Response to Chronic Exposure to High-Fat Diet and Pressure Overload"

_nutrients, 2023, doi:10.3390/nu15245139_

Round 1

Reviewer 1 Report

Comments and Suggestions for Authors

The manuscript presented by Tian et al deals with the role of TLR4 deficiency on cardiac function in pressure overload unter the influence of high-fat diet. 

The relevance/role of TLR4 deficiency has been analyzed previously. The only new aspect of the manuscript seems to be the long term data over 12 weeks.

The introduction is well written and the animal experiments are well depicted. 

The major concern is the novelty of the data, from the point of view that similar data have already been published. However, the results are coherent and consistent with the previous literature.

The labelling of the graphs is in the wrong ratio - it is advisable to readjust the labels after resizing from the original programme. 

Comments on the Quality of English Language

The manuscript presented by Tian et al deals with the role of TLR4 deficiency on cardiac function in pressure overload unter the influence of high-fat diet. 

The relevance/role of TLR4 deficiency has been analyzed previously. The only new aspect of the manuscript seems to be the long term data over 12 weeks.

Author Response

We thank reviewer 1 for all their positive comments regarding clarity of the massage and adequacy of our experimental planning. Furthermore, we revised and adjusted all figures along with their associated legends in our newly revised manuscript.

Reviewer 2 Report

Comments and Suggestions for Authors

The authors' study is very interesting and is conducted in an in vivo TLR4-/- model. The authors focus on the cardiovascular effects of a HF diet and in the presence of AAC. The data obtained are interesting but require some clarifications:

1- when it comes to hypertrophy, it would be appropriate to use not only histological but also molecular analysis by referring to markers known in the literature (BNP, ANF). Furthermore, an important data point from the point of view of cardiac functionality is always the pressure developed.

2- identify the pathway downstream of TLR4, therefore NF-kB and AP-1 and in particular the cytokines IL1b and caspase. It would therefore be appropriate to present the data also in light of these closely connected molecules.

3- it is not clear from a molecular point of view how the lack of TLR4 can determine these effects at the level of the response to two such powerful stress inducers.

Author Response

1- We thank reviewer 2 for  the compliment regarding the significance of our study. Furthermore, we would like to clarify that we did not check BNP or ANF levls as these mice were not showing any overt signs of heart failure or congestion.

2- We believe that the effect of HFD or AAC on heart function disturb the cardiac physiology (as evident y changes in EF and ESV but is not enough to cause congestion. Therefore, we believe that focusing on dynamic cardiac measures is more meaningful.

3- we agree with reviewer that exploring the effect of chronic HFD vs AAC exposure on downstream TLR4 signaling pathway is important. Therefore, we show expression of p65 and iKBb subcellular components of NF-kB. In addition, we added the expression of caspace-3 a downstream effector of NF-KB involved in apoptosis. New data and now shown in re-arranged Figure 2 and Figure 4.

Reviewer 3 Report

Comments and Suggestions for Authors

This manuscript uses Tlr4 knockout mice to show that Tlr4 acts pro-inflammatory or inhibitory to cardiac dysfunction when chronic stress of high-fat diet (HFD) causes cardiac hypertrophy and fibrosis, and chronic stress of abdominal arterial constriction (AAC) causes cardiac dysfunction, respectively. Of particular interest are the observation of a reduction in ejection fraction (EF%), a measure of cardiac function, and a significant increase in end-systolic volume (ESV) when Tlr4 knockout mice were chronically treated with AAC. The authors must respond to several comments below before publication.

1) Line 129: The notation for the mouse gene should be Tlr4, not TLR4. The same applies to the notation of other genes.

2) Line 139 (Section 3.1 title): HFD causes weight gain in TLR4 knockout mice as well as in WT (Fig. 1A, B)

3) Line 149: What does "p65 and IκBb subcellular components of NF-κB" mean?

4) Line 151-152: If inflammation caused by HFD is reduced in Tlr4 knockout mice, does this mean Tlr4 is involved in the inflammation caused by HFD in WT mice?

5) Line 172, Fig. 3H: EDV data not shown.

6) Line 195: Legends do not match with panels D, E, and F.

7) Lines 195, and 196: Is each panel in Fig. 4 reared in ND or in HFD?

8) Lines 206-208 (Fig. 6): quantitative data should also be shown.

Author Response

We appreciate reviewer 3 concerns and suggestion for improvement and correction of unintended mistakes. We addressed all the points raised by the reviewer regarding revising errors in nomenclature, clarifying graph labels and adding the missing panel in Figure 3.

Round 2

Reviewer 2 Report

Comments and Suggestions for Authors

I agree new version

Reviewer 3 Report

Comments and Suggestions for Authors

I have no more comment.